# Metabolic Alteration Bridging the Prediabetic State and Colorectal Cancer

**DOI:** 10.3390/cells13080663

**Published:** 2024-04-09

**Authors:** Antonino Colloca, Isabella Donisi, Camilla Anastasio, Maria Luisa Balestrieri, Nunzia D’Onofrio

**Affiliations:** Department of Precision Medicine, University of Campania Luigi Vanvitelli, Via L. De Crecchio 7, 80138 Naples, Italy; antonino.colloca@studenti.unicampania.it (A.C.); isabella.donisi@unicampania.it (I.D.); camilla.anastasio@unicampania.it (C.A.); marialuisa.balestrieri@unicampania.it (M.L.B.)

**Keywords:** metabolic alteration, prediabetes, colorectal cancer, chronic inflammation, obesity, hormone dysregulation, microbiota, nutrition, physical activity, pharmacotherapy

## Abstract

Prediabetes and colorectal cancer (CRC) represent compelling health burdens responsible for high mortality and morbidity rates, sharing several modifiable risk factors. It has been hypothesized that metabolic abnormalities linking prediabetes and CRC are hyperglycemia, hyperinsulinemia, and adipokines imbalance. The chronic stimulation related to these metabolic signatures can favor CRC onset and development, as well as negatively influence CRC prognosis. To date, the growing burden of prediabetes and CRC has generated a global interest in defining their epidemiological and molecular relationships. Therefore, a deeper knowledge of the metabolic impairment determinants is compelling to identify the pathological mechanisms promoting the onset of prediabetes and CRC. In this scenario, this review aims to provide a comprehensive overview on the metabolic alterations of prediabetes and CRC as well as an overview of recent preventive and therapeutic approaches for both diseases, focusing on the role of the metabolic state as a pivotal contributor to consider for the development of future preventive and therapeutic strategies.

## 1. Introduction

Prediabetes is a transition phase in the progression from a normal glucose tolerance to diabetes mellitus (DM), with high incidence and prevalence rates worldwide and most cases left untreated [1]. Three clinical criteria are applied for prediabetic diagnosis: (i) impaired glucose fasting (IGF), with fasting glycemic levels between 100 and 125 mg/dL; (ii) impaired glucose tolerance (IGT), with plasma glucose levels ranging from 140 and 199 mg/dL after oral glucose tolerance test (OGTT); and (iii) glycated hemoglobin (HbA1c) levels between 5.7% and 6.4% [1]. The global burden of prediabetes is rapidly growing in developed countries, with IGT and IGF prevalence percentages of 9.1% and 5.8%, respectively, in 2021, expected to increase by at least +1% by 2045 [2]. Population reports estimated that 70% of prediabetic patients will develop into type 2 DM patients, with 5% to 10% of prediabetic patients developing a clinical diabetic syndrome each year [3]. DM is a chronic hyperglycemia condition caused by impairment in insulin secretion, response to insulin, or both alterations [4,5], and is diagnosed by randomly measured glycemia ≥ 200 mg/dL, fasting glycemia ≥ 126 mg/dL, glycemic levels ≥ 200 mg/dL after a 2 h OGTT, and HbA1c ≥ 6.5% [6]. DM represents a heavy health burden, due to the high social and healthcare costs as well as its systemic complications, including chronic renal disease, diabetic retinopathy, non-traumatic limb amputations, heart failure, and increased cancer incidence and lethality [7,8,9]. Prediabetes results from the interaction of different factors, such as inflammatory and oxidative stress, obesity, and dysregulated hormonal pathways, which concur with the onset of metabolic impairment, insulin resistance (IR), and β-cell dysfunction [10]. Novel effective strategies targeting prediabetic metabolic damage would be pivotal tools to oppose prediabetic pathogenesis and prevent the development of other diseases characterized by severe metabolic alterations, including colorectal cancer (CRC). Indeed, evidence showed a correlation between the prediabetic and diabetic phenotype and the onset of CRC [9,11,12]. This neoplasia represents the third most diagnosed cancer worldwide, whose incidence and mortality rates are expected to increase over the years, particularly in developed countries, mostly linked to physical inactivity and unhealthy dietary habits, also related to T2DM and prediabetes [7,13,14]. Cell metabolic reprogramming during different stages of CRC carcinogenesis is associated with mutations in tumor suppressors and promoters, such as adenomatous polyposis coli (APC), Wnt, and MYC [15,16,17]. Several studies investigated the role of DM and prediabetes as independent risk factors for CRC development, identifying the existence of several diabetic pathways related to the immune system and metabolic regulation leading to CRC carcinogenesis [9,11,12,18]. Herein, an integrated and updated overview of the metabolic injuries characterizing prediabetes and CRC will be presented, as well as emerging preventive and therapeutic approaches targeting metabolic pathways in the pathogenesis of prediabetes and CRC.

## 2. Prediabetes: Risk Factors and Determinants

Insulin resistance (IR), characterized by impaired peripheral insulin activity, represents a critical state in prediabetes pathogenesis. The prediabetic insulin-resistant phenotype results from metabolic and inflammatory alterations, where complex interactions between genetic and environmental determinants play crucial roles [19,20]. Several factors, such as excessive nutrient intake, elevated dietary inflammatory index (DII), sedentary lifestyle, obesity, and psychological stress, can activate multiple pathways causing metabolic deregulation and decreased insulin sensitivity in insulin-respondent tissues [21,22] (Figure 1). Profound changes in pancreatic β-cells function and peripheral tissue insulin sensitivity, as well as increased inflammatory cytokines levels, are indicative of an altered incretin response and a chronic inflammatory state [23] (Figure 1). More recently, the contribution of the host–microbiome interaction, along with the alterations in gut microbiota, were unveiled as fundamental determinants in prediabetes onset and development [24] (Figure 1). In the following sections, the cornerstones of prediabetes pathogenesis will be extensively discussed, providing the basis for a better understanding of the relationship between prediabetes and CRC, and the definition of novel preventive and therapeutic strategies.

### 2.1. Chronic Inflammation

Inflammation and oxidative stress are crucial phenomena in the pathogenesis of prediabetes and, to date, the association between impaired glucose homeostasis and inflammation has been widely described [25,26]. Increased systemic inflammation assessed by upregulated inflammatory protein levels, such as resistin, interleukin (IL)-1β, tumor necrosis factor α (TNF-α), IL-6, monocyte chemoattractant protein-1 (MCP-1), and the hepatic marker C-reactive protein (CRP) to albumin ratio (CAR), has been reported in prediabetic patients [27,28]. Gonzale Delgado et al. described the fundamental role of inflammation in the pathogenesis of prediabetes, reporting that subjects with elevated inflammatory markers and a high body mass index (BMI) undergoing renal transplantation were more likely to develop prediabetes after kidney transplant [29]. Prediabetes is also characterized by systemic immune response dysregulation and inflammation in different tissues, such as the pancreatic islets and liver, pivotal regulators of glucose homeostasis. Indeed, prediabetic patients showed impaired immune response, with higher activation of the complement cascade, and hemostatic disorders, such as an increased production of coagulation factors [30]. Deregulation in the activity of cluster of differentiation (CD)+ T cells and of regulatory T cells (Treg), modulating T effector cell functionality, has been also reported in prediabetes. Particularly, Treg promoted T helper 17 differentiation and cytokine production in prediabetes, but not in DM, and showed overexpression of the fatty acid importer CD36, unveiling the critical role of host metabolome in controlling the immune response in prediabetes [31]. The environmental pollutant has emerged as a novel determinant in inflammation and the prediabetic state. Long-term exposure to air pollutants has been associated with the activation of inflammatory pathways and glycemic disorders proper of pediabetes [32], while the heavy metal cadmium reduced insulin secretion inducing β-cells death via ferroptosis [33]. The endocrine-disrupting compound bisphenol A triggered hypothalamic inflammation in a toll-like receptor 4 (TLR4)-dependent manner, promoting prediabetic metabolic dysfunction [34].

Several inflammatory-related microRNAs (miRNAs) were altered in the prediabetic state. Increased miR-27 and miR-195 levels related to the systemic inflammatory state and impaired insulin sensitivity have been assessed in the serum of obese prediabetic patients [35]. Similarly, in prediabetic subjects, downregulated expression of anti-inflammatory hsa-miR-146a-5p and upregulation of hsa-miR-1281 targeting hepatocyte nuclear factor 1 homeobox A (HFN1A), promoted hypoxia-inducible factor-1α (HIF-1α), vascular endothelial growth factor A (VEGFA), and vascular damage [36]. Overall, the pharmacological and lifestyle-based modulation of systemic inflammatory levels would impede the metabolic damage progression of prediabetes, as well as of CRC [37].

### 2.2. Obesity and Sarcopenia

Obesity is a pathological condition characterized by an excessive volume of adipose tissue, defined by a BMI value of higher than 30 kg/m2, and associated with several metabolic alterations [38]. Closely dependent on dysfunctional lifestyle habits in developed countries, obesity is a pandemic syndrome with high mortality and morbidity rates and is able to promote the onset of severe diseases [39]. The massive prevalence of obesity has been related to a concomitant increase in prediabetes [40]. An obesogenic diet results in hyperproliferation and altered differentiation of intestinal stem cells and progenitors, decreased serotonergic and increased peptidergic enteroendocrine cytotypes, and promotion of dysmetabolic pathways [41]. Abnormal nutrient intake represents a critical determinant in obesity, associated with a condition defined as “nutri-stress” where metabolic alterations induce an impaired heat shock proteins (HSPs) response, leading to mitochondrial damage, dysfunctional energy metabolism, high glucose levels, and IR [42]. Upon metabolic alterations, intestinal cells of prediabetic patients secrete higher levels of exosomal vesicles related to lipid metabolism and oxidative stress, compared to non-prediabetic subjects [43]. In obese patients, dysfunctional adipose tissue releases reactive oxygen species (ROS), inflammatory cytokines, and free fatty acids (FFAs), whose elevated plasmatic levels determine ectopic fat accumulation in non-adipose tissues [44]. Atherogenic alterations with increased free cholesterol, triglyceride, and saturated FFA levels have been also assessed in prediabetes patients, thus generating a lipotoxicity state with dysfunctional organelles and IR onset [44,45]. Evidence has suggested a pivotal role of mitochondrial dysfunction and endoplasmic reticulum stress in the development of IR conditions, with lipid and ROS accumulation [46]. A 2-week long high-fat diet was able to induce mitochondrial stress and acute IR in a mouse model [47], while fatty acid metabolites, such as diacylglycerols (DAGs), suppressed insulin signaling by activating protein kinase C(PKC)θ and PKCε, by IRS serine phosphorylation, and glucose transport inhibition [48]. The accumulation of ceramides altered mitochondrial chain function whilst sphingolipids led to PKCζ and protein phosphatase 2A (PP2A) activation, thus inhibiting protein kinase B (Akt)-induced glucose uptake [49]. Of note, the role of the mitochondrial guardian sirtuin (SIRT) 3 in insulin homeostasis and glucose and lipid metabolism has been reported in in vitro endothelial cells. Specifically, SIRT3 expression was associated with an enhancement of metabolic alteration induced by palmitic acid treatment and redox homeostasis [50]. Decreased SIRT6 and SIRT1 expression have been assessed in the abdominal fat of obese prediabetic patients, along with upregulated NF-κB, peroxisome proliferator-activated receptor gamma (PPAR-γ), and sterol regulatory element-binding transcription factor 1 (SREBP1) protein levels [51,52]. These results were corroborated by systemic inflammation, sustained by hyperglycemia and elevated CRP and cytokine content, such as IL-6 and TNF-α [51,52].

Obese patients display an increased expression of metallothionein (MT) 1 in pancreatic islets, which is negatively correlated with insulin secretin and β-cells failure [53]. Moreover, dysfunctional Langerhans islets and insulin secretion have been related to altered intercellular communications via Connexin36 (Cx36) gap junctions, which are downregulated in obesity and prediabetes [54]. In vivo and in vitro studies identified a correlation among miRNAs, obesity, and IR. Yu et al. described the ability of miR-27a to negatively regulate PPAR-γ expression in skeletal muscle cells, thus altering glycemic homeostasis [55]. The obesity-related sedentary life, along with dysregulated nutrition and excessive DII, are also associated with the progressive loss of muscular strength and sarcopenia of prediabetes, due to the fundamental role of myocytes in glucose homeostasis [56,57,58]. Grip strength and chair-rising time tests evaluated the impact of muscle strength on the metabolic alterations of prediabetic patients, correlating muscle strength to attenuated prediabetes evolution [59]. In addition, in prediabetic subjects, an increased DII score, defined by the individual intake of pro- and anti-inflammatory nutrients, is positively correlated with IR and reduced skeletal muscle mass and function [57]. In this scenario, raising awareness of the hurdles associated with poor nutrition and a sedentary lifestyle is pivotal to prevent the assessment of harmful metabolic reprogramming, characterizing prediabetes and CRC [37].

### 2.3. Hormonal Dysregulation

Along with insulin signaling, other deregulated hormonal pathways are involved in prediabetes pathogenesis [19,20]. The IGT condition of prediabetic patients has been correlated with higher hepatic levels of CD26/dipeptidyl peptidase 4 (DPP4) compared to normal glucose tolerant subjects, which determines massive degradation of glucagon-like peptide-1 (GLP-1) and glucose-dependent insulinotropic polypeptide (GIP), causing dysmetabolism and IR [60]. In addition, increased FFA levels reduce the expression of GLP-1 receptor and Cx-36, thus impairing insulin secretion [61]. Adiponectin and leptin are adipose tissue-secreted hormones regulating metabolic and inflammatory pathways at both peripheral and central levels [62]. Although in physiological conditions leptin ameliorates insulin sensitivity, its overexpression has been associated with fasting insulin plasma levels in obese and prediabetic patients, along with C1q/TNF-related protein 1 (CTRP1) [63]. The hepatokine fibroblast growth factor (FGF) 21 was also found elevated in obesity and hyperglycemia [64], representing a metabolic regulator of glucose and lipid homeostasis secreted by hepatocytes during nutritional stress [65]. A clinical study on prediabetic subjects assessed the upregulated FGF21/adiponectin ratio, which correlated with the onset and development of the condition [66]. Adiponectin and insulin-like growth factor binding proteins (IGFBP)-1 and IGFBP-2 levels directly reflect insulin sensitivity in adipose tissue and liver. Evidence evaluating the association between altered IGFBP levels and glucose homeostasis showed IGFBP-2 upregulation in women with pathological glucose tolerance, while increased IGFBP-1 levels were detected in male subjects [67]. Reduced adiponectin and nefastin-1 levels in prediabetes were associated with an increased risk of developing DM [68], whilst leucine-rich alpha-2-glycoprotein 1 (LRG1) expression was related to insulin dysmetabolism [69]. Indeed, overexpressed LRG1 was found in obese humans and mice, where it promoted hepatic steatosis, enhancing lipogenesis and suppressing fatty acid β-oxidation, and inhibited IRS1 and IRS2, thus promoting IR and prediabetes [69].

Thyroid hormones regulate glycemic homeostasis, modulating the insulin pathway response and affecting adipogenesis. A study involving 4378 patients showed a negative correlation between central thyroid hormone sensitivity, evaluated as an increased Thyroid Feedback Quantile-based Index (TFQI), TSH Index (TSHI), Thyrotrope Thyroxine Resistance Index (TT4RI), and prediabetes [70]. In addition, hypothyroid patients displayed increased circulatory leptin levels, suggesting its contribution to the development of IR, prediabetes, and DM [71].

Prediabetic subjects showed an altered insulin-antagonistic hormone axis with an enhanced responsivity to glucagon and cortisol and a reduced sensitivity to growth hormone (GH) [72]. The osteoblast-derived hormone lipocalin-2 is able to suppress appetite and reduce adipose tissue accumulation, improving glucose metabolism in obesity conditions [73]. Lipocalin-2 silencing results in a worse outcome in obese mice, while its increase improves insulin and glucose homeostasis, elucidating the physiological protective mechanisms of this hormone against prediabetes [73]. Moreover, increased plasmatic β-amyloid (Aβ)40 and Aβ42 levels have been related to impaired liver and muscle insulin sensitivity and pancreatic insulin secretion, fostering the onset of prediabetes [74]. Overall, the existence of a systemic interplay involving heterogeneous factors is emerging promptly as a crucial pathogenetic moment in the development of a dysmetabolic state. A better knowledge of the multifaceted effects exerted by hormones would be helpful for enabling the early diagnosis and strategical therapy of metabolic diseases.

### 2.4. Microbiota

The host–microbe relationship is promptly emerging as a crucial regulator of metabolic homeostasis and a critical regulator in prediabetic pathogenesis [75]. The alterations in microbiota diversity, often associated with unhealthy nutritional habits, can impair intestinal barrier integrity and permeability, leading to the state of endotoxemia, characterized by increased lipopolysaccharide translocation and a chronic state of inflammation [76]. Prediabetic patients show a deficiency in beneficial bacteria content, such as Lactobacillus and Bifidobacterium, accompanied by an increase in the content of proinflammatory bacteria. Patients display a reduced abundance of the mucin-degrading A. muciniphila as well as Clostridium bacteria, inversely correlated with fasting blood glucose and triacylglycerol levels, IR, inflammation, and obesity [75]. On the other hand, enhanced Dorea bacterial content, which is directly associated with glucose concentration, was assessed in prediabetic subjects [77].

Intestinal bacteria are able to produce different small-chain fatty acids (SCFAs), such as butyric acid, displaying a crucial role in inflammatory attenuation, as well as ameliorating insulin sensitivity [78]. Prediabetic subjects showed a significant reduction of the butyrate-producing bacterium Faecalibacterium prausnitzii, contributing to impaired glucose metabolism [79,80], while a decrease in Candidatus Soleaferrea and an accumulation of Parasutterella were associated with increased endotoxemia, chronic inflammation and IR [81].

In addition, a high-fat high-sugar diet was associated with changes in the intestinal microbiota with increased numbers of Erysipelotrichaceae bacteria, depleted effects of Th17 lymphocytes, and IL-17-mediated lipid uptake [82]. Evidence indicates the important role of microbiota in several diseases; furthermore, gut microbial dysfunctions have been also demonstrated to be directly involved in CRC pathogenesis [83]. To this end, the assessment of microbiota-targeting approaches could become an effective strategy against prediabetes, and even more so against CRC.

## 3. Prediabetes as an Independent Risk Factor for Colorectal Cancer

Prediabetes represents a transition phase in the passage from euglycemia to hyperglycemia and DM, thus resulting in associated risk factors for all DM-related mortality and morbidity events, such as cardiovascular diseases, dementia, and different tumors [84]. Indeed, glucose-intolerant patients are characterized by a higher total cancer risk compared to normal glucose-tolerant subjects [85], and prediabetic and diabetic subjects with tumors display an increased mortality rate [86,87]. Evidence demonstrated the association between prediabetes and CRC, sharing common negative lifestyle and environmental influences [21,88], including obesity, physical inactivity, dysregulated nutritional habits, microbiota alterations, and a metabolic reprogramming state [86,87]. Notably, it has been reported that the excessive consumption of red meat, and even more so if grilled or smoked, is an important risk factor for CRC, due to the formation of mutagenic and oxidative compounds during cooking processing and the alteration of the gut microbiota [89]. In this context, it should be noted that a high rate of consumption of meat is part of a form of ketogenic diet which is often followed by prediabetic patients and suggested by clinicians [90]. This treatment resulted in the normalization of blood glucose levels; however, following this diet for too long has been associated with toxic effects, i.e., an increase in cholesterol levels after six months [91].

As in the case of prediabetes, CRC is characterized by glucose dysmetabolism, with an accumulation of glycolytic intermediates and their diversion in different metabolic pathways, thus resulting in the increased production of lipids, amino acids, and nucleotidic molecules sustaining cell proliferation and survival [92]. Of note, evidence has suggested the direct influence of prediabetes in CRC development, the risk of which was enhanced in the advanced stages of prediabetes [93] (Figure 2). The impact of prediabetes, as well as DM, in CRC, can be ascribed to a chronic stimulation mediated by hyperglycemia, hyperinsulinemia, and hormonal dysregulations [86,94].

Mice affected by high-fat high-sugar diet (HFHSD)-induced prediabetes showed hyperproliferation, rapid differentiation, and rapid turnover of intestinal stem cells and progenitors, leading to an increased risk of cancer development [95]. These effects are dependent on upregulated PPAR-γ and SREBP1-mediated lipogenesis, inflammation, and cancer progression via activation of the pro-proliferative insulin receptor or insulin-like growth factor 1 receptor (IGF-1R)/Akt pathway [95]. An observational study supported the relationship between elevated fasting insulin and CRC risk, with increased HbA1c levels associated with CRC risk in men [96]. Hyperinsulinemia might also promote cell proliferation, invasion, and drug resistance in CRC [97], and IR, which is assessed as an elevated low-density lipoprotein (LDL)/high-density lipoprotein (HDL) ratio, and has been identified as a negative prognostic factor in CRC [98]. In vivo studies have described the overexpression of the insulin receptor, specifically its fetal isoform, in precancerous CRC lesions, supporting the role of insulin in cancer initiation and progression [99]. A close relationship between glycemia and CRC has also been assessed, with a relative risk of 1.015 per every 20 mg/dL increase in fasting plasma glucose [100], CRC-related mortality was associated with HbA1c levels, indicating that it plays a role in chronic glycemic alterations in the CRC phenotype [101]. Increased plasmatic glucose levels and nutrient availability result in increased ROS production determining metabolic reprogramming, as well as genetic and epigenetic alterations [102]. The activity of adipose tissue-derived factors, such as osteopontin, visfatin, and resistin, along with an imbalanced leptin/adiponectin ratio, promoted inflammation, CRC proliferation, and metastasis via integrin αvß6 expression [102,103,104]. A case-control study correlated enhanced resistin plasmatic levels to IR and CRC risk [105], while a clinical study on prediabetic subjects assessed the protective role of adiponectin against CRC by regulating TNF-α and VEGF levels [106]. On the contrary, reduced adiponectin expression was associated with colorectal polyp formation and malignant degeneration [107]. The role of prediabetes as an independent determinant in CRC pathogenesis is becoming clearer, concurring with the assessment of CRC as characterizing metabolic rearrangement and increasing cell proliferation and growth as well as favoring the acquisition of malignant features. To this end, the recognition of signs of the specific moments of prediabetes pathogenesis, such as insulin resistance, low-grade inflammation, altered hormone signaling, oxidative stress, and hyperglycemia, i.e., evaluating fasting glucose levels >100 md/dL or high sensibility CRP values near 2 mg/L, has great potentialities to become a novel bulwark against a CRC pandemic [108].

## 4. Common Therapeutic Approaches in Prediabetes and Colorectal Cancer

To date, the definition of effective strategies to prevent and treat prediabetes and CRC is crucial, given their severe health burden and poor outcomes. Different common approaches have been suggested for both diseases, such as acting on lifestyle habits, including dietary interventions, and the promotion of physical activity, as well as the off-label administration of antidiabetic drugs [109,110,111] (Figure 3). Here, an up-to-date overview of common strategies in prediabetes and CRC will be provided.

### 4.1. Nutrition

Unhealthy eating habits represent a leading cause in the onset and development of prediabetes and CRC [112,113], with increased DII as a critical parameter of both diseases [22,114]. At the same time, an adequate controlled nutritional approach can be a notable preventive strategy in the development of prediabetes and CRC [112,115].

Medical nutrition therapy is one of the most effective interventions, providing patients with a personalized dietary plan conceived by a clinical dietitian/nutritionist [116]. A recent trial evidenced the role of medical nutrition in the improvement of metabolic parameters, including FPG, HbA1c, insulin C-peptide, and cholesterol, in patients with prediabetes, DM, and high BMIs [117]. The effect of a hypocaloric ketogenic Mediterranean diet in rebalancing metabolic and anthropometric parameters has been reported and compared to a low-calorie non-ketogenic Mediterranean diet [117]. The ketogenic diet exerted beneficial effects suppressing CRC proliferation, via upregulation of the ketone body β-hydroxybutyrate and activation of hydroxycarboxylic acid receptor 2 (Hcar2)/homeodomain-only protein homeobox (Hopx) signaling [118]. In mice affected by high-fat diet-induced prediabetes, the calorie-restricting dietary regimen ameliorated glucose metabolism and Cx36 gap junction alterations, Ca2+-mediated mechanisms, and insulin secretion [54]. Similarly, in a CRC mouse model, caloric restriction was able to inhibit tumor growth and survival via upregulation of pro-apoptotic Bax, reduced Bcl2 and Ki67 levels, and restoration of CRC-induced gut dysbiosis [119]. Recent evidence described the role of food-derived bioactive compounds in different diseases, thus, their characterization could represent a critical strategy for a molecular-based nutritional approach for prediabetes and CRC [120,121]. The apple-derived phlorizin can be a competitive inhibitor of sodium-glucose co-trasporter-2 (SGLT2), ameliorating insulin sensitivity and reducing fecal microbiota-induced endotoxemia in obese prediabetic mice [122]. Moringa oleifera and ginseng supplementation improved glucose and lipid metabolic parameters, such as FPG, total cholesterol, HDL, and LDL profiles, in prediabetic subjects [123,124]. A randomized controlled trial showed that supplementation with red raspberries reduced total and LDL cholesterol, hepatic IR, and improved pancreatic β-cells function (NCT03049631) [125]. In palmitic acid-induced insulin-resistant endothelial cells, treatment with the dairy by-product whey induced beneficial effects on cell metabolism and the redox state [126]. Likewise, the consumption of phytonutrient-rich fruits and vegetables exerts chemopreventive effects in CRC patients, acting as antioxidant and anti-inflammatory compounds [127]. Tea-derived polyphenols opposed cell viability and proliferation in CRC, modulating the Wnt/β-catenin pathway [128], whilst delactosed milk whey (DMW) exerted chemopreventive activity by inducing in vitro apoptosis and restoring altered intestinal microbiota in a mouse model with azoxymethane-induced CRC [129]. The buffalo milk-derived δ-valerobetaine (δVB) exerted a pro-apoptotic effect in SW480 and SW620 CRC cells via PTEN-induced kinase 1 (PINK1)/Parkin pathway activation [130] and induced ROS-mediated apoptosis and SIRT6 upregulation in LoVo cells [131]. HCT116 and HT-29 cells treated with the milk-derived miR-27b underwent apoptotic cell death by mitochondrial ROS accumulation [132], while treatment with dietary-derived ergothioneine induced necroptotic death via activation of the SIRT3/Mixed Lineage Kinase Domain Like Pseudokinase (MLKL) pathway in CRC [133]. All this evidence highlights the importance of conscious nutrition as a first-line approach against metabolic diseases and emphasizes the role of bioactive compounds as epigenetic modulators with high potentialities in prediabetes and CRC, because of their target-specificity and low toxicity.

### 4.2. Physical Activity

Lack of exercise and a sedentary life are causes of a severe sanitary burden and are associated with chronic diseases, including prediabetes and CRC [134,135]. Promoting an active lifestyle led to decreased chronic disease-related mortality and morbidity [134], as exercise sessions improved glycemic homeostasis and insulin sensitivity in both healthy and glucose-intolerant subjects [136]. In responsive prediabetic patients, exercise training promoted a microbiome able to produce short-chain fatty acids and branched-chain amino acid degradation, while the non-responder microbiome mainly synthesized metabolically detrimental molecules [137]. Physical activity (PA) can modulate glucose homeostasis and affect metabolic parameters at different levels. Studies have demonstrated that PA downregulated leptin and IL-6 expression in prediabetic subjects [138], correlated with improved microbiota profile and reduced endotoxemia [139]. Randomized clinical trials (NCT02706262, NCT02706288) showed that regular exercise training, associated with a weight loss nutritional plan, was able to enhance metabolic benefits, with an increase in insulin sensitivity 2-fold higher in obese and prediabetic patients than when under a regimen of caloric restriction alone [140].

Recently, a meta-analysis evidenced the protective influence of moderate to high PA in digestive tract cancers [141], given the existence of a direct correlation between healthy lifestyle index, including PA, and CRC prevalence [142]. An increased exercise level was related to a reduction in CRC relative risk of up to 20% [143], as adequate PA is able to counteract CRC polygenic risk [144] and ameliorate the overall survival rate after surgery resection [145]. The exact mechanisms relating to CRC and PA are still unclear; however, it could be speculated that IR, chronic inflammation, and dysbiosis improvement represent key events of exercise-mediated cancer prevention [146,147]. In addition, PA could intervene in CRC natural pathogenesis affecting shear stress and opposing circulating tumor cell survival, regulating systemic milieu, reducing serum leptin, and increasing plasmatic adiponectin [148]. PA also decreased the levels of acidic and rich in cysteine (SPARC)-mediated apoptosis, induced the release of skeletal muscle anti-oncogenic extracellular vesicles, and promoted catecholamine-mediated immune cell mobilization [148].

Given the multiple beneficial effects exerted by PA, the promotion of an active lifestyle can be considered a first-line therapy, able to stop and revert the insulin resistance and dysmetabolism associated with prediabetes as well as to directly ameliorate CRC patient prognosis and life quality.

### 4.3. Pharmacotherapy

Antidiabetic drugs, including metformin, gliflozins, and incretin analogs, represent current therapeutic strategies for the prediabetic state, aimed at impairing the progression to DM and reducing the related morbidity and mortality rates (Table 1) [149,150]. However, this approach is not widely accepted as the progression from prediabetes to DM is not certain, underpinning doubtful outcomes of the risk/benefit and risk/cost ratios [151]. Treatment with metformin enhanced glycemic control counteracting DM progression in prediabetic patients [152,153] and, in addition to lifestyle changes, reduced DM evolution risk by 17% compared to lifestyle approach alone (NCT03441750) [154,155]. A randomized clinical study on prediabetic patients revealed the beneficial effects of metformin supplemented with probiotic Bifidobacterium treatment improving glucose homeostasis and opposing HbA1c and side effects, to a greater extent than single metformin treatment [156]. In different clinical and preclinical models of IR, exposure to metformin promoted insulin-dependent glucose transporter (GLUT) type 4 expression and its membrane translocation [157]. Obese prediabetic patients treated with metformin displayed lower systemic inflammation, oxidative stress, and pro-inflammatory miR-195 and miR-27 expression compared to non-treated subjects [158]. In addition, metformin increased SIRT6 expression and decreased inflammatory markers, including SGLT2, leptin, and the leptin/adiponectin ratio, in prediabetic patients with acute myocardial infarction, thus opposing coronary dysfunction, major adverse cardiac events, and prediabetic pericoronary fat accumulation [52,159,160]. SGLT2 inhibitors (SGLT2i) were also able to reduce cardiovascular death and heart failure-associated hospitalization in prediabetic patients [161]. Empagliflozin treatment ameliorated left ventricle reverse remodeling, enhancing patient ejection fraction [161], while a recent meta-analysis described the potentialities of SGLT2 inhibitors in preventing the evolution from prediabetes to DM [162]. However, it has been evidenced that a combination of SGLT2i treatment with a ketogenic diet exposes the patient to the risk of euglycemic ketoacidosis, contraindicating their combination in the treatment of prediabetic patients [163]. Similarly, GLP-1 receptor agonists (GLP1-RA) improved glucose homeostasis, and reduced body weight and systolic blood pressure, impairing the evolution of prediabetes to DM [164]. Treatment with GLP1-RA and GLP1-RA supplemented with GIP reduced body weight by 15% and 20% and decreased IR progression to DM [165], as the GLP1-RA liraglutide ameliorated IR and weight loss independently of GLP1-R signaling [166]. Liraglutide promoted insulin secretion, increasing the production of hippocampal cholinergic neurostimulating peptide (HCNP), choline acetyltransferase, and muscarinic receptor 3 (M3R) in prediabetic rat models [167], and, as in the case of other GLP1-RAs, improved glycemic control and thermogenesis and induced systemic and monocyte-derived IL-6 expression [168]. Exposure to liraglutide counteracted monocyte chemoattractant protein-1 (MCP-1) release, ameliorating inflammatory and atherosclerotic parameters [169], and promoted cardiovascular function by reducing tumorigenesis-2 (sST2) and troponin I inhibition (Eudract: 2013-001356-36) [170]. It is interesting to note that the protective effect of the GLP1-RA liraglutide alone on weight and visceral fat loss was lower than that of caloric restriction alone [171]. Recently, the interplay between gut microbiota and antidiabetic drugs, including metformin, SGLT2i, and GLP1-RA, has been extensively reviewed [172,173]. In detail, the gut flora composition, which affects metabolism and glucose homeostasis, can alter the efficacy of antidiabetic treatments [172,173]. On the other hand, metformin treatment can increase *Escherichia coli* and lower *Intestinibacter* content, also promoting the growth of several SCFA-producing beneficial bacteria, including *Blautia, Bacteroides, Butyricoccus, Bifidobacterium, Prevotella, Megasphaera,* and *Butyrivibrio* [172,173]. SGLT2i can stimulate beneficial *Alloprevotella Lactobacilli* spp. growth and reduce *Helicobacter* and *Mucispirillum* species prevalence [172,173]. GLP1-RA, such as liraglutide and dulaglutide, can increase the *Bacteroidetes* to *Firmicutes* ratio and the SCFA-producing *Bifidobacterium* content [172,173]

Several studies investigated the effects of current antidiabetic drugs on CRC (Table 1). Treatment with metformin led to increased overall and disease-free survival in CRC diabetic patients [174,175], opposing cell cycle and growth, cancer stem cells and metastatic ability via mammalian target of rapamycin (mTOR) and PI3K/Akt pathway inhibition and AMPK activation [176]. Metformin-derived antineoplastic effects are also mediated by suppression of tumor growth factor (TGF)-β/Inhibin Subunit βA (INHBA) signaling with downregulated cyclin D1 expression and cell cycle inhibition [177], along with urea cycle suppression and reduced putrescine levels [178]. A retrospective study in CRC diabetic patients reported a metformin-mediated survival benefit and reduced risk of liver metastasis after surgery [179]. Metformin induced caspase 3-independent apoptotic death in HCT116 and SW620 CRC cells [180] and disrupted the immunosuppressive effect of the tumor microenvironment [181]. More specifically, the antidiabetic drug exerted immunostimulating effects on CD8+ T lymphocytes following tryptophan metabolism reprogramming, which is reduced in CRC and increased in CD8+ [182]. In addition, metformin downregulated the mevalonate pathway as myeloid-derived suppressor cells and M2 macrophages in CRC mouse models [183] sensitized CRC cells to 5-Fluorouracil (5-FU) and irinotecan [184] and counteracted insulin-induced oxaliplatin resistance in HCT116 and LoVo cells via AMPK activation [97]. Evidence supports the possible role of different SGLT2is in CRC prevention and treatment. Dapagliflozin counteracted tumor growth, abrogating the hyperinsulinemia cancer-promoting effect in both in vivo and in vitro CRC models [185], and reducing cell adhesion, inducing loss of interaction with collagen I and IV associated with reduced Discoidin domain receptor family member 1 (DDR1) function in HCT116 cells not expressing the SGLT2 catabolizer UDP Glucuronosyltransferase Family 1 Member A9 (UGT1A9) [185]. A case report revealed a synergic interaction of SGLT2i and cetuximab in reducing metastatic CRC size and Carcinoembryonic Antigen (CEA) levels [186], whilst tofogliflozin suppressed CRC development and β-catenin accrual in diabetic mice [187]. In addition, evidence showed that SGLT2i reduced CRC by downregulating farnesylated Ras protein expression and plasmatic insulin levels [188]. An in vitro study indicated that GLP1-RA liraglutide suppressed CRC migration and survival, inducing apoptotic cell death through inhibition of the PI3K/Akt/mTOR pathway [189].

**Table 1 cells-13-00663-t001:** Effects of antidiabetic drugs on prediabetes and CRC.

Drug	Effects on Prediabetes	Effects on Colorectal Cancer
**Metformin**	Glucose homeostasis enhancement [152,153,155] Inhibition of prediabetes to DM progression [152,153] Increase in GLUT4 expression levels [157] Promotion of GLUT4 membrane translocation [157] Reduction in systemic inflammation and oxidative stress and miR-195 and miR-27 [158] SIRT6 upregulation [159] SGLT2 downregulation [159] Leptin/adiponectin ratio reduction [159]	Overall and disease-free CRC survival increase Reduced liver metastasis [174,175,179] Inhibition of mTOR and PI3K/Akt signaling [176,183] AMPK activation [97,176,183] TGF-β/INHBA signaling suppression [177] CyclinD downregulation [177] Urea cycle suppression [178] Reduced putrescin levels [178] Caspase 3-mediated apoptosis [180] Disruption of tumor-mediated immunosuppression [181,182,183] Chemosensitivity increase [97,184]
**Gliflozins**	Reduced cardiovascular-related death [161] Reduced heart failure [161] Inhibition of prediabetes to DM progression [162]	Suppression of hyperinsulinemia pro-tumoral effect [185] Reduced cell adhesion [185,186] Synergic cytotoxic effect with cetuximab [185] β-catenin suppression [187] Farnesylated Ras levels downregulation [188] Insulin levels downregulation [188]
**GLP-1RA**	Glucose homeostasis enhancement [164,168] Weight loss promotion [164,171] Amelioration of insulin sensitivity [166] Insulin secretion enhancement via HCNP, M3R [167] Choline acetyltransferase upregulation [167] Reduced systemic inflammation [168,169] Cardiovascular function enhancement [170]	Suppression of cell survival [189] Suppression of cell migration ability [189] Inhibition of PI3K/Akt/mTOR pathway [189] Reduced cancer relative risk [190]

Moreover, a recent cohort study described a protective role of GLP1-RAs against CRC in normal-weight diabetic patients and even more in obese diabetic patients [190]. Metformin, SGLT2i, GLP1-RA, and other antidiabetic drugs, by targeting some biological mechanisms, e.g., oxidative stress and mitochondrial dysfunction, demonstrated antioxidant and ROS scavenger properties in in vitro and in vivo models, a common feature of prediabetes and CRC [191,192]. Comprehensively, metformin, SGLT2i, and GLP1-RAs have been demonstrated to exert several beneficial effects in prediabetes, reducing inflammation and insulin resistance and inducing weight loss. Moreover, the off-label usage of antidiabetic drugs in CRC revealed new drug-specific mechanisms of action. To this end, further studies would allow us to add these new weapons to the therapeutic armamentarium against CRC.

## 5. Conclusions

Prediabetes and CRC globally represent severe health burdens given their high mortality and morbidity rates, thus, the definition of novel effective preventive and therapeutic strategies is compelling [2,13]. In this scenario, we provided an extensive overview of the current knowledge on prediabetes, evidencing its relevance not only in DM prevention but also as an independent disease with proper alterations and dysmetabolism [10]. Moreover, we highlight the strong relationship occurring between prediabetes and CRC, as this tumor is characterized by the accumulation of metabolic alterations during the progressive stages of carcinogenesis which promote cell proliferation and survival advantages [15,16,17,89]. Prediabetes and CRC display common risk factors, such as chronic inflammation, unhealthy lifestyle habits, and microbiota alterations, which ultimately concur with the establishment of global metabolic reprogramming [86,94]. Within this framework, overlapping lifestyle- and drug-based interventions have been investigated in prediabetes and CRC, supporting the existence of common pathological pathways in both diseases [109,110,111,193].

Above all, we provided evidence about the role of prediabetes as an independent determinant of CRC onset and progression, as prediabetic subjects have been characterized with higher CRC incidence and poorer prognosis compared to normal, glucose-tolerant subjects. However, the influence of prediabetes on CRC has been only partially explained, relying on the chronic state of hyperglycemia, hyperinsulinemia, and adipokine imbalance, which provide CRC cells with nutritional substrates and boost malignant phenotype acquisition [86,94]. In this scenario, further studies will allow us to exploit the potentialities of the prediabetes and CRC relationship and could result in a critical tool for designing novel effective targeted approaches aimed at counteracting systemic metabolic impairment and its clinical complications, and specifically preventing CRC onset and development.

## Figures and Tables

**Figure 1 cells-13-00663-f001:**
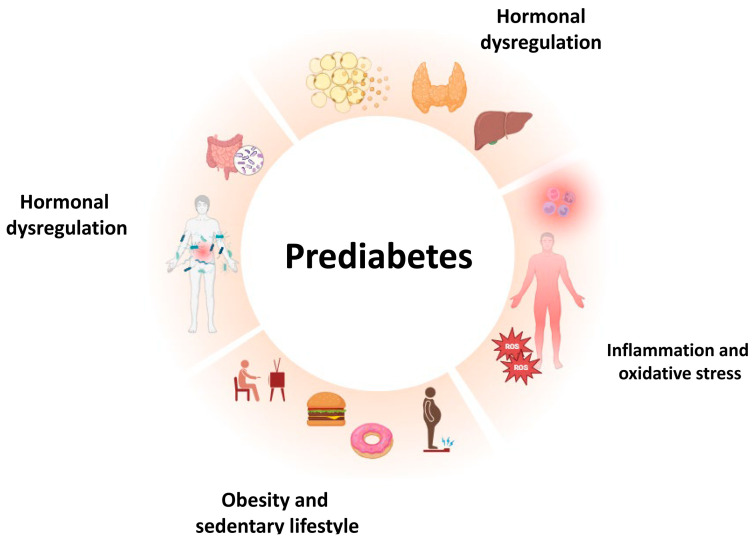
Risk factors and determinants in prediabetes pathogenesis. The interplay among different factors, including oxidative stress and inflammation, unhealthy lifestyle habits, hormonal dysregulation, and microbiota alterations, results in the aberrant activation of different metabolic pathways contributing to the pathogenesis of prediabetes.

**Figure 2 cells-13-00663-f002:**
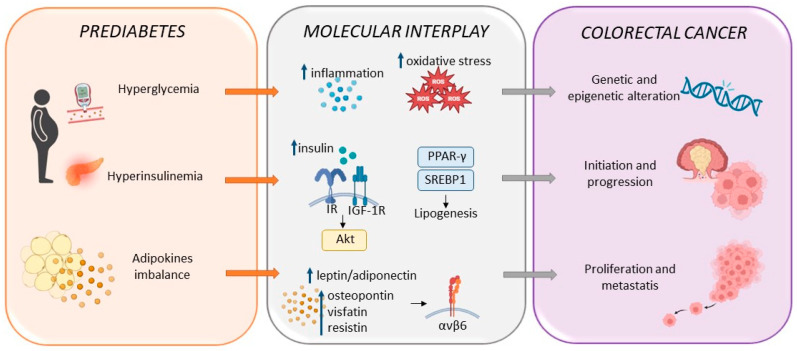
Prediabetes as an independent risk and prognostic factor in CRC. Schematic representation of typical prediabetes alterations and their molecular interplay as a trigger for CRC onset and development. PPAR-γ, peroxisome proliferator-activated receptor gamma; SREBP1 sterol regulatory element-binding transcription factor 1; IR, insulin receptor; IGF-1R, insulin-like growth factor 1 receptor; Akt, protein kinase B.

**Figure 3 cells-13-00663-f003:**
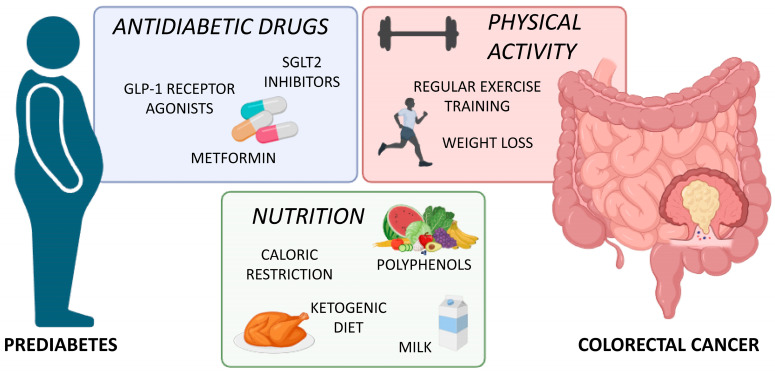
Common therapeutic strategies in prediabetes and CRC. A balanced nutritional plan rich in bioactive molecules accompanied by regular exercise sessions represents an effective approach against prediabetes and CRC. The off-label administration of antidiabetic drugs, such as metformin, GLP1 receptor agonists, and SGLT2 inhibitors, might also be effective in the prevention and treatment of both diseases. GLP1, glucagon-like peptide 1; SGLT2, sodium-glucose co-transporter 2.

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
