# Peer review of "Metabolic Alteration Bridging the Prediabetic State and Colorectal Cancer"

_cells, 2024, doi:10.3390/cells13080663_

Round 1

Reviewer 1 Report

Comments and Suggestions for Authors

The review manuscript submitted by Colloca et al. deals with two health burdens, prediabetic state and colorectal cancer (CRC), providing an up-to-date literature revision of risk factors behind the two diseases and aiming at deciphering the metabolic bridging between prediabetes and CRC.

Comments:

i)         The manuscript is very well written. It provides a very comprehensive and genuine review of the literature in relation to the two health burdens analysed and focuses on establishing metabolic paths for their bridging in order to define effective strategies to prevent CRC.

ii)        Advanced strategies provided by the authors are mostly metabolically driven and include nutritional interventions and adoption of an active lifestyle as well as some therapeutic approaches (e.g., antidiabetic drugs) that still require further validation. 

iii)      The authors also present prediabetes as an independent risk factor for CRC and provide a convincing reasoning for such.

Author Response

Response to reviewers’ comments

We thank the Reviewer for the helpful comments.

Reviewer 1: The review manuscript submitted by Colloca et al. deals with two health burdens, prediabetic state and colorectal cancer (CRC), providing an up-to-date literature revision of risk factors behind the two diseases and aiming at deciphering the metabolic bridging between prediabetes and CRC.

Comments:

  1. i)         The manuscript is very well written. It provides a very comprehensive and genuine review of the literature in relation to the two health burdens analysed and focuses on establishing metabolic paths for their bridging in order to define effective strategies to prevent CRC.
  2. ii)        Advanced strategies provided by the authors are mostly metabolically driven and include nutritional interventions and adoption of an active lifestyle as well as some therapeutic approaches (e.g., antidiabetic drugs) that still require further validation. 

iii)      The authors also present prediabetes as an independent risk factor for CRC and provide a convincing reasoning for such.

Reply: Authors deeply thank the Reviewer for the positive comments and critical understanding of the Manuscript.

Reviewer 2 Report

Comments and Suggestions for Authors

The present review is intriguing. The authors describe all molecular pathways hypothesized to be involved in the relationship between colon cancer and prediabetes—diabetes. Moreover, they suggest possible pharmaceutical treatment for prediabetes in terms of reducing the development of colon cancer.

The term ‘’plasmatic’’ in line 29 is wrong. I consider that the authors mean the ‘’plasma glucose levels’’. The term ‘’plasmatic’’ means fake.

The authors wrote about all modifiable risk factors for colon cancer development, including obesity, low physical activity, unhealthy diet, hormonal dysregulation, microbial dysbiosis, and dysmetabolism. According to the literature, eating grilled meat and food or/and barbecue, usually accompanied by vegetables, is an additional risk factor. Many patients with diabetes and prediabetes follow that type of diet, and it is a form of ketogenic diet. However, it is unhealthy when the patients follow that diet for a long time, although the blood glucose levels can be normalized. That diet could be used for a maximum period of six months, accompanied by supplements. So, the authors should add a paragraph focused on previously mentioned comments.

Moreover, the authors suggested a proposed treatment for the use of SGLT2i and the ketogenic diet. A combination of SGLT2i with the ketogenic diet is dangerous because of euglycemic ketoacidosis. The authors should add that limitation.

The manuscript is well written, and the discussion/conclusions are acceptable.

Overall, the data are of interest.

Comments on the Quality of English Language

none

Author Response

Response to reviewers’ comments

We thank the Reviewer for the helpful comments. We have addressed all issues according to the reviewer comments/suggestions. Changes are highlighted in green in the revised version of the manuscript.

Reviewer 2: The present review is intriguing. The authors describe all molecular pathways hypothesized to be involved in the relationship between colon cancer and prediabetes—diabetes. Moreover, they suggest possible pharmaceutical treatment for prediabetes in terms of reducing the development of colon cancer.

-The term ‘’plasmatic’’ in line 29 is wrong. I consider that the authors mean the ‘’plasma glucose levels’’. The term ‘’plasmatic’’ means fake.

Reply: We thank the Reviewer for the suggestion. According we corrected the term ‘’plasmatic’’ in “plasma glucose levels”. Please see line 30.

-The authors wrote about all modifiable risk factors for colon cancer development, including obesity, low physical activity, unhealthy diet, hormonal dysregulation, microbial dysbiosis, and dysmetabolism. According to the literature, eating grilled meat and food or/and barbecue, usually accompanied by vegetables, is an additional risk factor. Many patients with diabetes and prediabetes follow that type of diet, and it is a form of ketogenic diet. However, it is unhealthy when the patients follow that diet for a long time, although the blood glucose levels can be normalized. That diet could be used for a maximum period of six months, accompanied by supplements. So, the authors should add a paragraph focused on previously mentioned comments.

Reply: According to Reviewer suggestion, we discussed the effects of eating grilled meat and food or/and barbecue as additional CRC risk factor. “Notably, it has been reported that the excessive consumption of red meat, even more if grilled or smoked, is an important risk factor for CRC, given the formation of mutagenic and oxidative compounds during cooking processing and alteration of gut microbiota [89].” Please see lines 257-260 and new reference 89: Diakité, MT.; Diakité, B.; Koné, A.; Balam, S.; Fofana, D.; Diallo, D.; Kassogué, Y.; Traoré, CB.; Kamaté, B.; Ba, D.; Ly, M.; Ba, M.; Koné, B.; Maiga, AI.; Achenbach, C.; Holl, J.; Murphy, R.; Hou, L.; Maiga, M. Relationships between gut microbiota, red meat consumption and colorectal cancer. J Carcinog Mutagen. 2022;13(3):1000385.

In addition, as suggested in the Revised Manuscript the effects of ketogenic diet in patients with diabetes and prediabetes have been also provided. “In this context, it should be noted that the high assumption of meat is part of a form of ke-togenic diet, often followed by prediabetic patients and suggested by clinicians [90]. This treatment resulted effective in the normalization of blood glucose levels, nonetheless fol-lowing this diet for too long has been associated to toxic effects, i.e. an increase in choles-terol levels after six months [91]”. Please see lines 260-265, new reference 90: Bolla, AM.; Caretto, A.; Laurenzi, A.; Scavini, M.; Piemonti, L. Low-Carb and Ketogenic Diets in Type 1 and Type 2 Diabetes. Nutrients. 2019;11(5):962; and new reference 91: McGaugh, E.; Barthel, B. A Review of Ketogenic Diet and Lifestyle. Mo Med. 2022;119(1):84-88.

-Moreover, the authors suggested a proposed treatment for the use of SGLT2i and the ketogenic diet. A combination of SGLT2i with the ketogenic diet is dangerous because of euglycemic ketoacidosis. The authors should add that limitation.

Reply: As suggested, we provided additional and clear explanation concerning the limitation of effects SGLT2i during ketogenic diet. “However, it has been evidenced that a combination of SGLT2i treatment with a ketogenic diet exposes patient to the risk of euglygemic ketoacidosis, contraindicating their combi-nation in the treatment of prediabetic patients [163]”. Please see lines 433-436 and new reference 163: Mistry, S.; Eschler, DC. Euglycemic Diabetic Ketoacidosis Caused by SGLT2 Inhibitors and a Ketogenic Diet: A Case Series and Review of Literature. AACE Clin Case Rep. 2020;7(1):17-19.

-The manuscript is well written, and the discussion/conclusions are acceptable. Overall, the data are of interest.

Reply: Authors deeply thank the Reviewer for the positive comments and critical understanding of the Manuscript.

Reviewer 3 Report

Comments and Suggestions for Authors

The manuscript # cells-2947611 focusing on metabolic alterations that are common for the prediabetic state and colorectal cancer, is a topic of time. The paper is  well written and organized, but I would like to pay attention of Authors to some issues that were not sufficiently presented:

1. The backgroud of the Abstract is too extensive. It lacks:

-the major findings, namely the discription of metabolic state(abnormalities) linking prediabetes and CRC,

-conclusions,

-clinical aspect (how changes in metabolic state may affect the risk of CRC development).  

2. In the Chapter 2.4  the mechanisms of intestinal dysbiosis induded endotoxemia and  inflammation should be preceisly explained.

3. On Figure 2, there is the interplay between prediabetes and CRC, try to use arrows to show the directions of the interactions e.g.  hyperglycemia and hyperinsulinemia (in prediabetes, blue) results from disturbances in insulin pathway (yellow ) leading to increase in glycolysis (purple). What is the source of ROS, what represents yellow and blue circles, do alteration in lipogenesis rate derived from insulin signaling dysfunction and  hyperglycemia and hyperinsulinemia? Gut  dysbiosis is a common feature of prediabtes and CRC, but this is not clear from the figure. Does inflammation result from gut dysbiosis? The font size of scheme from glucose to puryvate should be larger.

4. At the end of Chapter 3 you wrote " To this end, the identification of  effective preventive and therapeutic approches in prediabetes, focusing on its specific pathogenetic moments, has great potentialities to become a novel bulwark against CRC  pandemic.". Try to clearly indicate what events  can be consideresed as "specific pathogenetic moments". Can they be the first signs of insulin resistance, low grade inflammation, oxidative stress? What clinical parameters (and thier value) should be taken into account by clinicians in order to recommend physical activity, changes diet habits and pharmacotherapy?

5. Please note that antidiabetic drugs, including metformin, gliflozins and incretin analogues reduce hyperglycemia-induced oxidative stress, which is also one of the common feature of prediabetes and CRC. It should be added to the Chapter 5.2. 

6. Do antidiabetic drugs (pharmacotherapy) alone affect gut dysbiosis or  biodiversity of gut microbiome?

Comments on the Quality of English Language

I found some minor language errors such as in Figure 2 hyperglicemia instead of hyperglycemia. Thus I recommend to go through the whole paper to search for typos. 

Author Response

Response to reviewers’ comments

We thank the Reviewer for the helpful comments. We have addressed all issues according to the reviewer comments/suggestions. Changes are highlighted in blue in the revised version of the manuscript.

Reviewer 3: The manuscript # cells-2947611 focusing on metabolic alterations that are common for the prediabetic state and colorectal cancer, is a topic of time. The paper is well written and organized, but I would like to pay attention of Authors to some issues that were not sufficiently presented:

  1. The backgroud of the Abstract is too extensive. It lacks:

-the major findings, namely the discription of metabolic state (abnormalities) linking prediabetes and CRC,

-conclusions,

-clinical aspect (how changes in metabolic state may affect the risk of CRC development). 

Reply: Authors deeply thank the Reviewer for the critical comments. As suggested we rephrased the abstract section, including the abnormalities linking prediabetes and CRC, how changes in metabolic state may affect the risk of CRC development and conclusions. “Prediabetes and colorectal cancer (CRC) represent compelling health burdens responsible of high mortality and morbidity rates, sharing several modifiable risk factors. It has been hypothesized that metabolic abnormalities linking prediabetes and CRC are hyperglycemia, hyperinsulinemia and adipokines imbalance. The chronic stimulation related to these metabolic signatures can favor CRC onset and development, as well as negatively influenced CRC prognosis. To date, the growing burden of prediabetes and CRC has generated a global interest in defining their epidemiological and molecular relationships. Therefore, a deeper knowledge of the metabolic impairment determinants is compelling to identify the pathological mechanisms promoting the onset of prediabetes and CRC. In this scenario, this review aims at providing a comprehensive overview on the metabolic alterations of prediabetes and CRC as well as an overview on recent preventive and therapeutic approaches for both diseases, focusing on the role of the metabolic state as a pivotal contributor to consider for the development of future preventive and therapeutic strategies”. Please see lines 10-21.

  1. In the Chapter 2.4 the mechanisms of intestinal dysbiosis induded endotoxemia and inflammation should be preceisly explained.

Reply: According to Reviewer observations, in the chapter 2.4 the mechanisms of intestinal dysbiosis induced endotoxemia and inflammation have been precisely explained. “The alterations in microbiota diversity, often associated to unhealthy nutritional habits can impair intestinal barrier integrity and permeability, leading to the state of endotoxemia, characterized by increased lipopolysaccharide translocation and a chronic state of inflammation [76]”. Please see lines 225-228 and new reference 76: Di Tommaso, N.; Gasbarrini, A.; Ponziani, FR. Intestinal Barrier in Human Health and Disease. Int J Environ Res Public Health. 2021, 18(23):12836.

  1. On Figure 2, there is the interplay between prediabetes and CRC, try to use arrows to show the directions of the interactions e.g. hyperglycemia and hyperinsulinemia (in prediabetes, blue) results from disturbances in insulin pathway (yellow) leading to increase in glycolysis (purple). What is the source of ROS, what represents yellow and blue circles, do alteration in lipogenesis rate derived from insulin signaling dysfunction and hyperglycemia and hyperinsulinemia? Gut dysbiosis is a common feature of prediabtes and CRC, but this is not clear from the figure. Does inflammation result from gut dysbiosis? The font size of scheme from glucose to puryvate should be larger.

Reply: As suggested, we rearranged Figure 2 in a clearer manner, using the appropriate arrows to explain the relationships among the different factors. Please see new Figure 2.

  1. At the end of Chapter 3 you wrote " To this end, the identification of effective preventive and therapeutic approches in prediabetes, focusing on its specific pathogenetic moments, has great potentialities to become a novel bulwark against CRC pandemic." Try to clearly indicate what events can be consideresed as "specific pathogenetic moments". Can they be the first signs of insulin resistance, low grade inflammation, oxidative stress? What clinical parameters (and thier value) should be taken into account by clinicians in order to recommend physical activity, changes diet habits and pharmacotherapy?

Reply: According to Reviewer observations, at the end of Chapter 3 we provided a clear indication about the mentioned "specific pathogenetic moments", adding accounting clinical parameters. “To this end, the recognition of signs of the specific moments of prediabetes pathogenesis, such as insulin resistance, low-grade inflammation, altered hormone signaling, oxidative stress and hyperglycemia, i.e. evaluating fasting glucose levels >100 md/dl or high sensibility CRP value near 2 mg/L, has great potentialities to become a novel bulwark against CRC pandemic [108].” Please see lines 308-312 and new reference 108: Ghule, A.; Kamble, TK.; Talwar, D.; Kumar, S.; Acharya, S.; Wanjari, A.; Gaidhane, SA.; Agrawal, S. Association of Serum High Sensitivity C-Reactive Protein With Pre-diabetes in Rural Population: A Two-Year Cross-Sectional Study. Cureus. 2021;13(10):e19088.

  1. Please note that antidiabetic drugs, including metformin, gliflozins and incretin analogues reduce hyperglycemia-induced oxidative stress, which is also one of the common feature of prediabetes and CRC. It should be added to the Chapter 5.2.

Reply: According to Reviewer comment, the role of antidiabetic drugs, including metformin, gliflozins and incretin analogues in reducing hyperglycemia-induced oxidative stress, which is also one of the common feature of prediabetes and CRC has been added. “Metformin, SGLT2i, GLP1-RA and other antidiabetic drugs by targeting some biological mechanisms e.g., oxidative stress and mitochondrial dysfunction demonstrated antioxidant and ROS scavenger properties in vitro and in vivo models, a common feature of pre-diabetes and CRC [191,192]”. Please see lines 489-492, new reference 191: Teodoro, JS.; Nunes, S.; Rolo, AP.; Reis, F.; Palmeira, CM. Therapeutic Options Targeting Oxidative Stress, Mitochondrial Dysfunction and Inflammation to Hinder the Progression of Vascular Complications of Diabetes. Front Physiol. 2019; 9:1857; and new reference 192:  Bardelčíková, A.; Šoltys, J.; Mojžiš J. Oxidative Stress, Inflammation and Colorectal Cancer: An Overview. Antioxidants (Ba-sel). 2023;12(4):901.

  1. Do antidiabetic drugs (pharmacotherapy) alone affect gut dysbiosis or biodiversity of gut microbiome?

Reply: According to Reviewer observations, we provided additional informations about the effects of antidiabetic drugs and their interplay with gut microbiota. “Recently, the interplay between gut microbiota and antidiabetic drugs, including metformin, SGLT2i and GLP1-RA, has been extensively reviewed [172,173]. In detail, gut flora composition, affecting metabolism and glucose homeostasis, can alter the efficacy of antdiabetic treaments [172,173]. On the other hand, metformin treatment can increase Escherichia coli and lower Intestinibacter content, also promoting the growth of several SCFAs producing beneficial bacteria, including Blautia, Bacteroides, Butyricoccus, Bifidobacterium, Prevotella, Megasphaera and Butyrivibrio [172,173]. SGLT2i can stimulate beneficial Alloprevotella Lactobacilli spp. growth and reduce Helicobacter and Mucispirillum species prevalence [172,173]. GLP1-RA, such as liraglutide and dulaglutide, can increase the Bacteroidetes to Firmicutes ratio and SCFA-producing Bifidobacterium content [172,173]”. Please see lines 450-460, new reference 172: Kant, R.; Chandra, L.; Verma, V.; Nain, P.; Bello, D.; Patel, S.; Ala, S.; Chandra, R.; Antony, MA. Gut microbiota interactions with anti-diabetic medications and pathogenesis of type 2 diabetes mellitus. World J Methodol. 2022;12(4):246-257, and new reference 173: Liu, W.; Luo, Z.; Zhou, J.; Sun, B. Gut Microbiota and Antidiabetic Drugs: Perspectives of Personalized Treatment in Type 2 Diabetes Mellitus. Front Cell Infect Microbiol. 2022; 12:853771.

Round 2

Reviewer 2 Report

Comments and Suggestions for Authors

The authors answered to my comments

Comments on the Quality of English Language

none

Reviewer 3 Report

Comments and Suggestions for Authors

Your responses to my comments and changes in the manuscrpit are sufficient.